# Grip Strength Trajectories and Cognition in English and Chilean Older Adults: A Cross-Cohort Study

**DOI:** 10.3390/jpm12081230

**Published:** 2022-07-27

**Authors:** Bárbara Angel, Olesya Ajnakina, Cecilia Albala, Lydia Lera, Carlos Márquez, Leona Leipold, Avri Bilovich, Richard Dobson, Rebecca Bendayan

**Affiliations:** 1Public Health Nutrition Unit, Institute of Nutrition and Food Technology, University of Chile, Santiago 7830490, Chile; bangel@inta.uchile.cl (B.A.); llera@inta.uchile.cl (L.L.); cmarquez@inta.uchile.cl (C.M.); 2Department of Biostatistics and Health Informatics, Institute of Psychiatry, Psychology and Neuroscience, King’s College London, London WC2R 2LS, UK; olesya.ajnakina@kcl.ac.uk (O.A.); leipold.leona@kcl.ac.uk (L.L.); richard.j.dobson@kcl.ac.uk (R.D.); rebecca.bendayan@kcl.ac.uk (R.B.); 3Department of Behavioural Science and Health, Institute of Epidemiology and Health Care, University College London, London WC1E 6BT, UK; 4Latin Division, Keiser University eCampus, Fort Lauderdale, FL 33409, USA; 5NIHR Biomedical Research Centre at South London and Maudsley, NHS Foundation Trust and King’s College London, London SE5 8AF, UK; 6Centre for the Study of Decision-Making Uncertainty, University College London, London WC1E 6BT, UK; a.bilovich@ucl.ac.uk; 7Health Data Research UK London, University College London, London WC1E 6BT, UK; 8Institute of Health Informatics, University College London, London WC1E 6BT, UK; 9NIHR Biomedical Research Centre, University College London Hospitals NHS Foundation Trust, London NW1 2PG, UK

**Keywords:** grip strength, cognition, older adults, longitudinal study

## Abstract

Growing evidence about the link between cognitive and physical decline suggests the early changes in physical functioning as a potential biomarker for cognitive impairment. Thus, we compared grip-strength trajectories over 12–16 years in three groups classified according to their cognitive status (two stable patterns, normal and impaired cognitive performance, and a declining pattern) in two representative UK and Chilean older adult samples. The samples consisted of 7069 UK (ELSA) and 1363 Chilean participants (ALEXANDROS). Linear Mixed models were performed. Adjustments included socio-demographics and health variables. The Declined and Impaired group had significantly lower grip-strength at baseline when compared to the Non-Impaired. In ELSA, the Declined and Impaired showed a faster decline in their grip strength compared to the Non-Impaired group but differences disappeared in the fully adjusted models. In ALEXANDROS, the differences were only found between the Declined and Non-Impaired and they were partially attenuated by covariates. Our study provides robust evidence of the association between grip strength and cognitive performance and how socio-economic factors might be key to understanding this association and their variability across countries. This has implications for future epidemiological research, as hand-grip strength measurements have the potential to be used as an indicator of cognitive performance.

## 1. Introduction

In the last decade, our understanding of ageing—a multifactorial process involving complex interactions between key biomarkers such as cognitive and physical functioning—has increased markedly. Both cognitive and physical decline are a universally accepted part of normative ageing [1,2,3,4], with growing evidence that the two are inextricably linked [5,6,7]. Leisman et al. have argued that there is a dynamic bidirectionally link between motor and cognitive processes due to the shared neural pathways in the brain [1]. An increasing number of studies suggest that impaired physical performance precedes cognitive decline, thus highlighting early changes in physical functioning as a potential biomarker for cognitive impairment and dementia [8,9,10,11,12,13,14,15].

A number of physical function tests have been explored as biomarkers for cognition, including dual-task gait, gait speed, chair rise time and handgrip strength, with evidence suggesting that different measures of physical functioning are associated with different cognitive domains [2,3,4,5,6,7,8]. Handgrip strength has emerged as an indispensable biomarker for both current status and future outcomes of health in older adults [9,10,11,12,13]. The reduction in handgrip strength is considered to have a predictive value in relation to health markers, and this ability to predict adverse events has led to its use as a marker of sarcopenia and musculoskeletal function index [14,15]. The versatility of this measure, as well as the ease with which it can be assessed in clinical settings [16], has spearheaded the interest in handgrip strength as a biomarker for cognitive decline.

Numerous studies have documented the cross-sectional relationship between handgrip strength and cognitive function in later life [17,18,19,20], unanimously reporting a positive correlation between the two. While this association has not been found in healthy young (20–30 years old) adults [16], it becomes apparent in middle adulthood and increases with age; weak handgrip strength has been associated with 1.35 times higher odds of mild cognitive impairment (MCI) in those aged 50 to 64 years old compared to 1.54 times higher odds in those aged 65 or older [21].

Given the clear association in cross-sectional studies, longitudinal studies investigating the temporal and causative relationship between grip strength and cognitive function have gained significant traction in recent years [22,23,24,25,26,27,28,29,30,31,32,33,34,35,36,37,38,39,40,41,42,43,44,45,46,47,48,49,50,51,52,53,54]. The majority of these studies investigate the rate of change of cognitive function or grip strength, relative to baseline grip strength or cognitive function, respectively, with a much smaller number examining the relationship between grip strength and cognition over time [35,36,37,45,51,52,53]. Of note, is that the vast majority of these longitudinal studies focus on high income countries, with very few exploring this association within populations in the Global South [43,54,55].

Despite the growing body of work, results remain ambiguous, with no clear pattern emerging between studies (perhaps unsurprisingly, given the inherent complexities involve in these studies, with multiple additional factors affecting both handgrip strength and cognitive function [56,57,58,59,60,61]). A meta-analysis by Cui et al. revealed poorer grip strength was associated with cognitive decline and onset of dementia [62], whilst a second meta-analysis found only lower limb function, and not grip strength, to be associated with an increased risk of developing dementia [6]. Other systematic reviews have concluded there is little evidence of longitudinal associations among rates of change [63], in contrast to the findings by Fritz et al. that support the use of handgrip strength measurements over time as a tool to predict cognitive decline [64].

Given the current landscape of mixed results, there is a clear need for further longitudinal studies that examine the relationship between the rates of change of cognitive function and handgrips strength in representative populations in the global south. Therefore, the main aim of this study is to explore the association between cognitive status and physical function (i.e., grip strength) in older adults from representative samples in Chile and the UK. Both countries have longitudinal studies in community-dwelling older people (ELSA and ALEXANDROS), with comparable measurements, similar dates of evaluations and years of follow-up [65,66]. Furthermore, Chile and the UK have a life expectancy at birth of over 80y and similar life expectancy at 65y (UK 20y; Chile 19.9y) [67], but in different cultural and socioeconomic contexts, healthcare systems and demographic dynamics, which is more valuable the comparison. More specifically, we want to examine whether there are any significant changes in physical function in individuals as a function of identified changes in cognitive status over the years of follow-up. We hypothesize that individuals which were already cognitive impaired at baseline will have decreased physical function at baseline when compared to those that were not cognitively impaired at baseline and that those that have developed cognitive impairment over the follow-up years will show the fastest decline in all physical function over the follow up years.

## 2. Materials and Methods

### 2.1. Setting and Sample

The English Longitudinal Study of Ageing (ELSA) is an ongoing large, multidisciplinary study of a nationally representative sample of English adults aged ≥50 years. The ELSA study started in 2002 (wave 1), with participants recruited from an annual cross-sectional survey of households who were then followed-up every 2 years. Refreshment samples are recruited periodically to ensure that the full age range remains fully represented. Compared with the national census, the ELSA sample is representative of the non-institutionalized general population aged ≥50 in the UK [65]. To date, there have been nine waves of data collection spanning a follow-up period of 18 years, providing detailed information on health, lifestyle and socioeconomic circumstances for each ELSA participant. For the present study, baseline data were obtained from wave 2 (2004–2005) for the core members who started at wave 1 with information on the grip strength available at wave 2 (2004–2005), wave 4 (2008–2009), wave 6 (2012–2013) and wave 8 (2016–2017)—a follow-up period of 12 years. We excluded participants with a diagnosed organic cause of cognitive decline, such as dementia, at baseline. There were no significant differences between our final analytical sample and those respondents who were excluded for relevant variables in this study. ELSA received ethical approval from the South Central—Berkshire Research Ethics Committee (21/SC/0030, 22 March 2021).

The ALEXANDROS cohort, designed to study disability and dementia associated with obesity in community dwelling people 60y and older, has already been described [66]. Briefly, Alexandros include three cohorts: (a) the SABE study—a multi-center study conducted in seven Latin American capitals (PAHO), comparable to ELSA, established in 1999–2000, (b) The ALEXANDROS cohort recruited in 2005 and 2008 randomly selected from the Public Primary health Care Centers, and (c) the ISAPRES cohort that includes people of high socioeconomic status. The cohort consists of 3086 individuals (2880 of which were randomly recruited from the registers of health centers). After approval by the Institutional Scientific Ethics Review Board of the Institute of Nutrition and Food Technology (INTA) of the University of Chile (Acta n°23, 2012, FONDECYT n°1130947), and informed consent signature were performed. Of these, *n* = 1670 participants completed questionnaires about general information, sociodemographic background, history of chronic diseases, cognitive evaluation test, quality of life, physical activity and self-perception of health and lifestyles among others. In addition, complete anthropometric evaluations were undertaken, including mobility and physical performance tests. The ALEXANDROS cohort had three waves of data collection spanning a 16-year follow-up period. For the present study, baseline data were obtained from Wave 1 (2000–2005) for the primary members who started in Wave 1 with information on hand grip strength available in Wave 2 (2008–2011) and Wave 3 (2013–2016). Total sample sizes in both the ELSA and ALEXANDROS cohorts are presented in Table 1.

### 2.2. Variables

Grip Strength. In ELSA, grip strength of the dominant hand was assessed using the Smedley hand-held dynamometer (Stoelting Co., Wood Dale, IL, USA); participants were required to hold the device at a right angle to their body and exert maximum force for a couple of seconds when instructed. The average of three measurements was used as the final measure of the grip strength. In ALEXANDROS, the hand grip strength was measured with a T-18 manual dynamometer (Country Technology, Inc., Gays Mills, WI, USA) with an accuracy of 0.1 kg in subjects with reference measurements before 2008 or with the JAMAR brand for measurements made from 2008 onwards; these measurements were made according to the Southampton protocol with previously calibrated dynamometers, using the dominant hand and recording the highest value after two measurements. 

Cognitive status. In ELSA, the scores from immediate and delayed recall tests were used as a measure of cognitive status. Respondents were asked to recall as many words as possible from a list of 10 common nouns immediately after the list was read and after a short delay of 5 min, during which they completed other cognitive tests. Total scores are the number of words recalled in each test and ranged between 0 and 10, with higher scores indicative of better memory and cognitive status. We used a cut-off of ±2SD to distinguish individuals that were cognitively normal vs. those that were cognitively impaired. In ALEXANDROS, the scores of four questions included in the Mini-mental State Examination (MMSE) short version were used as a measure of cognitive status. Three of these questions related to immediate and late memory (indicate current date, repeat three objects aloud and remember them again after a few minutes) in addition to one question related to executive function (Digits Forward and Backward subtests (WAIS-R or WAIS-III)). Total scores ranged from 0 to 15, with higher scores indicative of better memory and cognitive status. Once again, the ±2SD cut-off was used to distinguish cognitively normal vs. cognitively impaired individuals. When we compared this definition of cognitive status with the Gold Standard (full MMSE), the results in older people in the ALEXANDROS cohort showed 85.6% sensitivity and 96.3% specificity.

For both the ELSA and ALEXANDROS cohorts, we identified three main groups based on their cognitive performance over the follow-up period: (1) the NON-IMPAIRED group included individuals who were not cognitive impaired at baseline and the last wave (Wave 8 in ELSA and Wave 3 in ALEXANDROS); (2) the IMPAIRED group encompassed individuals who were cognitively impaired at baseline and remained impaired in the last wave considered; and (3) the DECLINED group included participants who were not cognitively impaired at baseline but they were identified as cognitively impaired in the last wave of each study.

Covariates. Age, sex, Body Mass Index (BMI), education, wealth, self-rated health (SRH), Activities of Daily Living (ADL), smoking (current smoker and non-smoker was a reference), physical activity (physically active and physically inactive was a reference) and depressive symptoms were included as covariates in association analyses. 

Depressive symptoms were measured with an 8-item version of the Centre for Epidemiologic Studies Depression Scale [68,69], which has been found to have comparable psychometric properties to the full 20-item scale; a score ≥ 4 was used to define participants with severe depressive symptoms [70,71]. Functioning was measured with ADLs which were used to derive a binary variable to capture no difficulties and difficulties. Smoking status was assessed by asking if respondents have ever smoked and was coded as a binary variable (0 = never and 1 = ever smoked) [72]. Physical activity was assessed with three questions that asked participants how often they took part in activities that were either vigorous (e.g., jogging, cycling), moderately energetic (e.g., gardening, walking at moderate pace) or mildly energetic (e.g., laundry, home repairs); response options were: more than once a week, once a week, 1–3 times a month, hardly ever/never. Following other studies, responses to these questions were used to create a new physical activity variable which had four categories (0 = inactive, 1 = mild physical activity, 2 = moderate physical activity, 3 = vigorous physical activity) [73]. 

In ALEXANDROS, the Geriatric Depression Scale test (GDS-15 [74,75]) was used to assess depressive symptoms. On this scale a normal score is 0–4 points and depressive symptoms ≥ 5 points.

In both the ELSA and ALEXANDROS cohort, the following variables were defined and measured in the same way. For BMI [weight (kg)/height (m^2^)], body weight was measured by a study nurse using electronic scales without shoes and in light clothing, and height was measured using a Stadiometer. Physical inactivity was defined as being inactive on a weekly basis and/or only mild activities, that included doing laundry or home repairs only at least once a week [73]. Self-rated health was measured by an item asking the respondents about their health perception. Responses were grouped as 0 = poor or fair, and 1 = good, very good or excellent, following previous studies [76,77]. Educational attainment was measured with how many years of schooling each participant completed.

### 2.3. Statistical Analyses

Group differences were examined using chi-square, ANOVA and Kruskal–Wallis tests.

To examine the grip strength trajectories over time, linear mixed models (LMMs) with random coefficients [78] were estimated. LMMs have been shown to be useful for the analysis of longitudinal data with an evitable loss to follow-up data [79]. Linear unconditional and conditional models were fitted to ease comparability and an unstructured covariance structure was assumed. BIC estimates were used as fit statistics and lower BIC indicate better fit. For each outcome, adjustments were included in four steps: Model 1 shows unadjusted models (only including our cognitive grouping variable); Model 2 is adjusted for socio-demographics such as age, gender and educational attainment; and Model 3 was additionally adjusted for health status (comorbid health conditions including nutritional status, depression and limitations in ADLs) and health-related behaviors, such as smoking and levels of physical activity. All association analyses were conducted in STATA release 15 (StataCorp LLC, Lakeway Drive, TX, USA).

Imputation of missing values. In the ELSA study, the proportion of missingness in the covariates ranged from 0 (age, gender, depressive symptoms) to 35.4% (BMI); assuming the missing values were dependent on observed and not unobserved data [80], we imputed the missing values employing missForest in RStudio version 3.6.2 [80]. MissForest is an iterative imputation method based on Random Forests that handles continuous and categorical variables equally well and accommodates non-linear relation structures [80]. MissForest has been shown to outperform well-known imputation methods, such as *k*-nearest neighbors and parametric multivariate imputation by chained equations, in complex datasets that included non-linearities [80]. To evaluate the quality of imputation, we estimated the imputation error using Normalized Root Mean Squared Error (NRMSE) for continuous variables and proportion of falsely classified (PFC) for categorical variables [80]. A value close to 0 represents an excellent performance, and a value of 1 indicates poor performance. In ELSA, the imputation of the missing values yielded a minimal error (NRMSE = 0.07%; PFC = 0.10%), highlighting that the imputed values were very closely aligned with the observed values for both continuous and categorical variables.

Missing data in covariates were estimated in ALEXANDROS. In the ALEXANDROS study, the proportion of missingness in the covariates ranged from 0 (age, gender, self-rated health) to 5.03% (smoke). Assuming the missing values were missing completely at random (MCAR), we imputed the missing values employing multiple imputation analysis in STATA 15.

## 3. Results

### 3.1. Descriptive Statistics

Baseline sample characteristics of ELSA and ALEXANDROS cohorts are provided in Table 1. The maximal sample at baseline in ELSA comprised *n* = 7486 with an average age of 66.5 years old (SD = 9.7); of the entire sample, 45.4% were men, 26.5% reported to have poor health and 14.4% were smokers. The three primary groups encompassed Non-Impaired (*n* = 5180, 69.2%), Declined (*n* = 608, 8.1%) and Impaired (*n* = 1698, 22.8%). Individuals categorized as Non-Impaired were more likely to be women (56.6%), younger (mean = 64.1, SD = 8.5), have more years of education (mean = 14.0, SD = 3.8) and lower levels of depression (12.6%) compared to participants encompassed in non-impaired and impaired groups. The Declined group (*n* = 608) was younger (mean = 67.7, SD = 9.5) and only slightly more highly educated than the Impaired group (mean = 12.6 vs. mean = 11.8). When compared with those in the Impaired group, individuals in the Declined group were more likely to have a better perception of their health (67.9% vs. 60.1%), report no difficulties with activities of daily living (77.5% vs. 71.6%) and less likely to be depressed (16.2% vs. 21.0%). 

For the ALEXANDROS cohort, the sample comprised *n* = 1363 participants with an average age of 66.9 years old (SD = 4.7 years); of the entire sample, 29.5% were men with an average schooling of 8 ± 4.7 years. Only 10% of the entire cohort self-reported their health status as poor, 10.2% reported depressive symptoms and 9.5% indicated that they were a recurrent smoker. The three primary groups encompassed Non-Impaired (*n* = 1161, 85.2%), Declined (*n* = 118, 8.6%) and Impaired (*n* = 84, 6.2%). The participants who were included in the Non-Impaired group were younger (mean = 66.7 years old, SD = 4.5) than the Declined (mean = 68.5 years old, SD = 5.4) and Impaired (mean = 68.6 years old, SD = 6.2) groups. The impaired group presented a higher frequency of difficulties with activities of daily living (26.5%) and depressive symptoms (22.4%) when compared to the non-impaired group (2.7% and 8.5%, respectively). Regarding physical activity, it is noteworthy that only 6.7% and 6.4% of them indicate that they perform high and moderate physical activity, respectively.

#### 3.1.1. Physical Functioning Outcomes

Distribution of the physical functioning outcomes across different waves in the ELSA cohort is provided in Table 2 and for the ALEXANDROS cohort in Table 3. At the start of the study, the average grip strength in the entire ELSA cohort was 29.2 (SD = 11.5). Compared to the Impaired group, the Non-Impaired group had the highest score on grip strength (mean = 30.3, SD = 11.4) followed by the Declined group (mean = 28.9, SD = 11.8); the Impaired group had the lowest measure of grip strength (mean = 25.9, SD = 11.2) (F = 93.5, df = 7485, *p* < 0.001). This trend in distribution of the grip strength measures persisted in the following waves of data collections. Similar results were observed in the ALEXANDROS cohort where the highest score on grip strength was obtained for the non-impaired group (mean = 24.3, SD = 9.6) followed by Declined (mean = 20.5, SD = 7.5) and Impaired groups (mean = 20.8, SD = 8.8) (F = 12.3, df = 1362, *p* = 0.002). The same pattern of results was observed in the following waves of data collection.

#### 3.1.2. Linear Mixed Models

The results from the associations between cognitive impairment and longitudinal measure of grip strength in older adults within each cohort are provided in Table 4. In the ELSA cohort, at intercept level, grip strength scores (baseline wave for each study) in the Declined (Model 1: β = −1.45, 95%CI = −2.40–−0.50, *p* = 0.003) and Impaired (Model 1: β = −3.83, 95%CI = −4.47–−3.19, *p* < 0.001) groups were significantly lower than the Non-Impaired group in our unadjusted model. The difference between the Impaired and Non-impaired group remained significant and was only partially attenuated when adjusting for socio-demographics, health status and health behaviours (Model 2: β = −1.06, 95%CI = −1.53–−0.60, *p* < 0.001; Model 3: β = −0.54, 95%CI = −1.01–−0.08, *p* = 0.022). However, the difference between Declined and Non-Impaired was fully attenuated after adjustments for socio-demographics (Model 2: β = −0.07, 95%CI = −0.71–0.57, *p* = 0.826; Model 3: β = 0.15, 95%CI = −0.49–0.79, *p* = 0.639). At slope level, we found that individuals in the Declined (Model 1: β = −0.17, 95%CI = −0.27–−0.07, *p* = 0.001) and Impaired (Model 1: β = −0.17, 95%CI = −0.30–−0.04, *p* = 0.010) groups have a significantly faster decline in their grip strength over time when compared to those individuals in the Non-Impaired group. However, these differences became non-significant when adjustments for socio-demographics, health status and health behaviours were considered.

In ALEXANDROS, at intercept level, our results showed that the Declined (Model 1: β = −3.93, 95%CI = −5.69–−2.18, *p* < 0.001) and Impaired (Model 1: β = −3.43, 95%CI = −5.54–−1.32, *p* < 0.001) group had significantly lower grip strength scores when compared to the Non-Impaired. While the relationship between grip strength scores and the Impaired group became non-significant when adjusting for socio-demographics, health status and health behaviours (Model 3: β = −1.44, 95%CI = −3.11–0.24, *p* = 0.093), the association between lower grip strength scores at baseline and the Declined group (Model 1: β = −3.93, 95%CI = −5.69–−2.18, *p* < 0.001) remained significant in the fully adjusted model (Model 3: β = −1.69, 95%CI = −2.96–−0.42, *p* = 0.009). At slope level, only those in the Declined group had a significantly faster decline in their grip strength over time (Model 1: β = −0.16, 95%CI = −0.30–−0.02, *p* = 0.024), and this was only partly attenuated in the fully adjusted model (Model 3: β = −0.15, 95%CI = −0.29–−0.004, *p* = 0.044). There were no significant differences between the Impaired group when compared to the Non-Impaired group across all three models.

## 4. Discussion

The main aim of this paper was to examine the association between cognitive and physical function over 12–16 years in two representative samples of UK and Chilean older adults. We compared grip strength trajectories over time in three groups of individuals based on their cognitive performance at baseline and last follow-up for each study. The group labelled as Non-Impaired represents those individuals who were not cognitively impaired; the group labelled as Impaired consisted of those individuals who had cognitive impairment at baseline and remained impaired in the last wave considered; the last group labelled as Declined included those that were not cognitively impaired at baseline but became cognitively impaired in the last wave. For ELSA, the differences found between the Impaired group and the Non-Impaired group remained after considering socio-demographics, health status and health behaviours. However, the differences between the Declined group and the Non-Impaired group were fully attenuated when considering potential confounders. In contrast, for ALEXANDROS, the differences found between the Non-Impaired group and the declined group were the only ones that remained significant after considering potential confounders. At slope level, those that declined (for both populations) or were impaired (for ELSA) showed a significantly faster decline in their grip strength over time when compared to those individuals in the non-impaired group; however, these tend to be fully or partially attenuated by socio-demographics, health status and health behaviours.

These findings are consistent with previous research showing the association between cognitive impairment and motor function in general [22,23,34,81,82,83,84], and grip strength in particular [17,18,19,20]. Similar results were found in a study conducted on a representative sample of Colombian people, showing a strong inverse association between handgrip strength and different domains of intrinsic capacity, including cognition [55]. This is in keeping with the meta-analysis carried out by Rijk et al., analyzing the predictive value of handgrip strength in cognition, mobility, functional status and mortality in older community-dwelling people, showing the association between low handgrip strength and decline in cognition [12]. A second meta-analysis by Cui et al. revealed similar results [62], although the review conducted by Kueper et al. showed only the association of incident dementia with lower limb function, which can be explained by the inclusion of people with Parkinson disease in the sample [6]. These results, in conjunction with the availability and ease of collection of grip strength data, further support the adequacy of using grip strength as a relevant indicator in cognitive ageing research and as a valuable measure of muscle function—as indicated by its inclusion in the sarcopenia diagnostic algorithm of the European Working Group on Sarcopenia in Older People [15].

With regard to the comparison of the grip strength trajectories between groups, we found that the Declined and Impaired group show a faster decline in their grip strength over time when compared to those individuals in the Non-Impaired group. Similarly, studies of Korean [36], Swedish [45], Canadian [52] and Dutch [53] populations have found associations between rates of change in cognition and handgrip strength, in line with the Fritz et al. review, which showed that the decrease in handgrip strength is a predictor of cognitive loss [64]. However, there are also a number of reviews and studies that find little evidence of longitudinal associations among rates of change [35,36,51,63]; the associations among rates of change found in this work were fully explained by socio-demographics, health status and health behaviours in ELSA and partially attenuated in ALEXANDROS. This highlights the key role that covariates play in this complex relationship, with the differential impact of potential confounders potentially due to differences in socio-demographics, such as education. Chilean participants had lower levels of education when compared with ELSA participants. Moreover, the Chilean cohort had, on average, baseline dynamometry measurements much lower than that of the English. 

To the best of our knowledge, this study is the first to compare grip strength trajectories across three patterns of change in cognitive status between a European and a South American country. Not only do we address a major epidemiological question which is key to our understanding of healthy ageing in both populations, but we also address the urgent need for international cross-cohort replication and reproducible research. Recent publications have highlighted the need to promote systematic replication efforts, especially in longitudinal studies of aging [85]. Moreover, an integrative systematic data analysis approach was used (Hofer and Piccinin, 2009). As Hofer highlights, another possible source of variation when performing replications or comparing results between studies is the different statistical procedures followed. In addition, it allows the comparison of two very different realities. Although Chile has the longest life expectancy at birth and the highest GDP per capita in South America, the socioeconomic situation is markedly different from that in England. A clear example is the lower educational attainment of Chilean older people, which, in turn, is linked to health and socio-economic inequalities that are clearly reflected in the Gini Coefficients for Chile and the UK in 2017 (44.4 and 35.1, respectively). Our study highlights the need to promote cross-national comparisons including non-European/North American countries (which is currently the most common approach), particularly considering the important modulating role of socio-economic factors on the association between physical function and cognitive function in older adults.

Some limitations should be also acknowledged. Although we have performed sensitivity analyses to account for mortality, our findings might represent the subset of healthy survivors which is a common limitation in longitudinal studies of ageing [86]. Although the results from these two representative survey samples of the UK and Chile contribute to improving our understanding of the role of socio-economic factors on the association between grip strength and cognitive performance from a public health perspective, future replications with other countries would be of great interest to detangle the impact of education inequalities on these ageing phenotypes. Furthermore, although we cross-validated the cognitive measure used in ALEXANDROS with MMSE cut-off points, some variability in our results could have been associated with these different measurements, and future studies should include other cognitive measures or specific cognitive domains. Finally, causal relationships cannot be attributed, and future research should aim to further examine the directionality of these relationships.

## 5. Conclusions

To sum up, our study provides robust evidence of the association between grip strength and cognitive performance and how socio-economic factors might be key to understanding this association and their variability across countries. This has direct implications for future epidemiological research, as hand grip strength measurements are easy to apply and compare. As such, it could be valuable in countries with different socioeconomic backgrounds, where it has the potential to be used as an indicator of cognitive performance when this may not be so easily assessed.

## Figures and Tables

**Table 1 jpm-12-01230-t001:** Baseline sample characteristics of ELSA and ALEXANDROS cohorts.

	ELSA Cohort				ALEXANDROS Cohort			
	Total	Non-Impaired	Declined	Impaired	Total	Non-Impaired	Declined	Impaired
	*n* = 7486	*n* = 5180	*n* = 608	*n* = 1698	*n* = 1363	*n* = 1161	*n* = 118	*n* = 84
	Mean (SD)/*n* (%)	Mean (SD)/*n* (%)	Mean (SD)/*n* (%)	Mean (SD)/*n* (%)	Mean (SD)/*n* (%)	Mean (SD)/*n* (%)	Mean (SD)/*n* (%)	Mean (SD)/*n* (%)
**Socio-demographic characteristics**								
Age (years)	66.5 (9.7) *	64.1 (8.5)	68.2 (9.6) ^ab^	73.0 (9.9) ^ab^	66.9 (4.7) *	66.7 (4.5)	68.5 (5.4) ^a^	68.6 (6.2) ^a^
Sex (male)	3397 (45.4) **	2246 (43.4)	301 (49.5) ^a^	850 (50.1) ^a^	402 (29.5)	356 (30.7)	25(21.2)	21 (25.0)
Education (years)	13.4 (3.7) *	14.0 (3.8)	12.6 (3.5) ^ab^	11.8 (3.1) ^ab^	8.0 (4.7) *	8.5 (4.6)	5.4 (3.9) ^ab^	4.1 (3.6) ^ab^
**Comorbid health issues**								
Poor self-rated health	1986 (26.5) **	1114 (21.5)	195 (32.1) ^ab^	677 (39.9) ^ab^	136 (10.0) ***	126 (10.9)	7 (5.9) ^a^	3 (3.6) ^a^
BMI (kg)	27.9 (4.8)	27.9 (4.8)	28.1 (4.5)	27.7 (4.8)	28.8 (4.9)	28.8 (4.8)	29.2 (5.4)	28.7 (4.6)
ADL (≥1)	1457 (19.5) **	838 (16.2)	137 (22.5) ^ab^	482 (28.4) ^ab^	59 (4.3) **	31 (2.67)	6 (5.1)	22 (26.5) ^a^
Depression (score ≥ 5)	1095 (14.8) **	646 (12.6)	99 (16.3) ^ab^	350 (21.0) ^ab^	115 (10.2) **	82 (8.5)	18 (17.5)	15 (22.4)
**Behavioral outcomes**								
Currently a smoker	1078 (14.4)	748 (14.4)	82 (13.5)	248 (14.6)	125 (9.5) ***	108 (9.6)	13 (11.1)	4 (5.1)
Physical activity	**				**			
High	1141 (20.6)	881 (22.2)	94 (21.3)	166 (14.6)	90 (6.7)	85 (7.3)	2 (1.7)	3 (3.6)
Moderate	2930 (52.9)	2170 (54.8)	224 (50.8)	536 (47.1)	87 (6.4)	84 (6.9)	1 (0)	4 (4.8)
Low	1468 (26.5)	909 (22.9)	123 (27.9) ^b^	436 (28.3) ^ab^	1178 (86.9)	1048 (86.6)	116 (98.3) ^a^	76 (91.6)

SD, standard deviation; ADLs, activities of daily living. * Fisher exact test *p* < 0.001, ** Test x^2^ *p* < 0.001, *** Test x^2^ *p* < 0.05; ^a^ *p* < 0.05 comparison *v*/*s* Non-impaired, (Bonferroni/Tests of proportions); ^b^ *p* < 0.05 comparison Decline *v*/*s* Impaired (Bonferroni).

**Table 2 jpm-12-01230-t002:** Distribution of the physical functioning outcomes across different waves in ELSA cohort.

	ELSA (*n* = 7486)			
	Wave 2 *	Wave 4 *	Wave 6 *	Wave 8 **
Grip Strength	Mean (SD)/*n* (%)	Mean (SD)/*n* (%)	Mean (SD)/*n* (%)	Mean (SD)/*n* (%)
Total Sample (*n* = 7486)	29.2 (11.5)	28.1 (11.3)	26.8 (10.6)	26.5 (10.3)
Non-Impaired (*n* = 5180)	30.3 (11.4)	28.8 (11.3)	27.4 (10.6)	26.8 (10.3)
Declined (*n* = 608)	28.9 (11.8)	27.7 (11.6)	26.0 (11.0)	26.4 (11.1)
Impaired (*n* = 1698)	25.9 (11.2)	25.2 (10.9)	24.2 (10.2)	24.4 (9.9)

SD, standard deviation. * Fisher Exact Test *p* < 0.001, ** Fisher Exact Test *p* = 0.002.

**Table 3 jpm-12-01230-t003:** Distribution of the physical functioning outcomes across different wave in the ALEXANDROS cohort.

	ALEXANDROS (*n* = 1363)		
	Wave 1 *	Wave 2 *	Wave 3 *
Grip Strength	Mean (SD)/*n* (%)	Mean (SD)/*n* (%)	Mean (SD)/*n* (%)
Total Sample (*n* = 1363)	23.7 (9.5)	24.2(9.4)	22.9 (8.5)
Non-Impaired (*n* = 1161)	24.3 (9.6)	24.7 (9.3)	23.7 (8.6)
Declined (*n* = 118)	20.5 (7.5) **	19.9 (9.1)	19.2 (6.6) *
Impaired (*n* = 84)	20.8 (8.8) **	21.3 (9.7)	20.3 (8.3) *

SD, standard deviation. * Fisher Exact Test *p* < 0.001, ** Fisher Exact Test *p* = 0.002.

**Table 4 jpm-12-01230-t004:** Linear mixed models for grip strength trajectories for ELSA and ALEXANDROS.

	Model 1			Model 2			Model 3		
ELSA Cohort	β	95%CI	*p*-Value	β	95%CI	*p*-Value	β	95%CI	*p*-Value
**Baseline**									
Non-Impaired	-	-	-	-	-	-	-	-	-
Declined	−1.45	−2.40–−0.50	0.003	−0.07	−0.71–0.57	0.826	0.15	−0.49–0.79	0.639
Impaired	−3.83	−4.47–−3.19	<0.001	−1.06	−1.53–−0.60	<0.001	−0.54	−1.01–−0.08	0.022
**Rate of change**									
Non-Impaired	-	-	-	-	-	-	-	-	-
Declined	−0.17	−0.27–−0.07	0.001	−0.09	−0.20–0.01	0.087	−0.10	−0.21–0.01	0.075
Impaired	−0.17	−0.30–−0.04	0.010	−0.10	−0.23–0.04	0.167	−0.12	−0.25–0.02	0.095
BIC		129,159.80			103,165.00			96,695.58	
-LL(model)		−64,530.64			−51,485.56			−48,184.00	
**Variance ^a^**									
Within-person	0.19	0.14–0.27		0.13	0.08–0.22		0.11	0.06–0.20	
In initial status	113.68	109.66–117.86		33.94	32.24–35.73		30.14	28.54–31.84	
In rate of change	−2.06	−2.47–−1.64		−0.47	−0.75–−0.20		−0.32	−0.59–−0.06	
**ALEXANDROS cohort Baseline**									
Non-Impaired	-	-	-	-	-	-	-	-	-
Declined	−3.93	−5.69–−2.18	<0.001	−1.53	−2.78–−0.29	0.016	−1.69	−2.96–−0.42	0.009
Impaired	−3.43	−5.54–−1.32	<0.001	−1.02	−2.54–0.49	0.185	−1.44	−3.11–0.24	0.093
**Rate of change**									
Non-Impaired	-	-	-	-	-	-	-	-	-
Declined	−0.16	−0.30–−0.02	0.024	−0.16	−0.30–−0.01	0.030	−0.15	−0.29–−0.004	0.044
Impaired	−0.04	−0.22–0.15	0.683	−0.08	−0.27–0.11	0.421	−0.04	−0.25–0.17	0.682
BIC		21,082.31			19,470.85			18,403.59	
-LL(model)		−10,500.94			−9719.427			−9172.794	
**Variance ^a^**									
Within-person	0.04	0.01–0.21		0.04	0.01–0.19		0.03	0.003–0.28	
In initial status	67.39	61.02–74.41		22.07	18.91–25.77		21.30	18.04–25.16	
In rate of change	−0.77	−1.26–−0.29		−0.57	−0.93–−0.20		−0.51	−0.90–−0.13	

CI, confidence interval; BIC, Bayesian information criterion, ^a^ the within-person variance is the overall residual variance in cognition that is not explained by the model. The initial status variance component is the variance of individuals’ intercepts about the intercept of the average person. The rate of change variance component is the variance of individual slopes about the slope of the average person. Model 1 is unadjusted model. Model 2 is adjusted for age, gender and educational attainments. Model 3 in addition to covariates included in Model 2 are further adjusted for comorbid health conditions, such as body mass index, depression and total number of impaired activities of daily living and health-related behaviours, such as smoking and levels of physical activity.

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
