# Peer review of "Grip Strength Trajectories and Cognition in English and Chilean Older Adults: A Cross-Cohort Study"

_jpm, 2022, doi:10.3390/jpm12081230_

Round 1
Reviewer 1 Report
Perhaps there is no need to separate descriptive from linear statistics, since it is only stated "Descriptive statistics. Group differences were examined using chi-square, ANOVA 211 and Kruskal–Wallis tests." If you decide to keep the lines, try to be more specific in defining why was each test selected or for which variables.
When adjusting by depression, which instrument was used?
When mentioning (weight (kg)/height (m2 )), one parenthesis could be changed to bracket.
When adressing educational attainment as the years an individual has completed, is it a reliable variable? In certain contexts schoolyears may or may not really reflect attainment. Please provide references.
When measuring self-perceived health, was it an open question? commonly it is defined by a period of time, in general, overall, in the last year, in the last mont
Author Response
We appreciate all comments and suggestions made by the reviewer, the changes made are in the text marked in yellow. One of the authors, who is native English-speaking, reviewed the manuscript again. Here are our responses to his questions or comments:
Reviewer 1
1) Perhaps there is no need to separate descriptive from linear statistics, since it is only stated "Descriptive statistics. Group differences were examined using chi-square, ANOVA 211 and Kruskal–Wallis tests." If you decide to keep the lines, try to be more specific in defining why was each test selected or for which variables.
R: We appreciate the reviewer's comment, this suggestion was incorporated and the descriptive and linear statistics are presented together in the text.
2) When adjusting by depression, which instrument was used?
R. To adjust for depression, this variable was used for the presence of depressive symptoms, according to each of the instruments used.
3) When mentioning (weight (kg)/height (m2)), one parenthesis could be changed to bracket.
R. We appreciate the suggestion and it was incorporated into the text by: [weight (kg)/height (m2)]
4) When adressing educational attainment as the years an individual has completed, is it a reliable variable? In certain contexts, schoolyears may or may not really reflect attainment. Please provide references.
R. For our study, using years of formal education was the best way to standardize education between the cohorts, since they are different systems for both populations. Below we give you an account of some previous publications, where the variable years of education has been used.
- Albala C, Sánchez H, Lera L, Angel B, Cea X. Efecto sobre la salud de las desigualdades socioeconómicas en el adulto mayor: Resultados basales del estudio expectativa de vida saludable y discapacidad relacionada con la obesidad (Alexandros) [Socioeconomic inequalities in active life expectancy and disability related to obesity among older people]. Rev Med Chil. 2011 Oct;139(10):1276-85. Spanish. Epub 2012 Jan 3. PMID: 22286726.
- Fuentes P, Albala C. An update on aging and dementia in Chile. Dement Neuropsychol. 2014 Oct-Dec;8(4):317-322. doi: 10.1590/S1980-57642014DN84000003. PMID: 29213920; PMCID: PMC5619178.
- Lövdén M, Fratiglioni L, Glymour MM, Lindenberger U, Tucker-Drob EM. Education and Cognitive Functioning Across the Life Span. Psychol Sci Public Interest. 2020 Aug;21(1):6-41. doi: 10.1177/1529100620920576. PMID: 32772803; PMCID: PMC7425377.
5) When measuring self-perceived health, was it an open question? commonly it is defined by a period of time, in general, overall, in the last year, in the last mont.
R. We appreciate the reviewer's comment, when we ask about self-perception of health, it is done at the time of the interview (currently) and it is not an open question, since a Likert scale is used for the answers:
How is your health in general?
1)Excellent 2) Very good 3) Good 4) Fair 5) or,poor

Reviewer 2 Report
Dear Authors,
an interesting issue, thank you!
I have just some small remarks:
1) please, give 1-2 sentences in the Introduction (in the beginning) that ELSA and ALEXANDROS can be compared; this is slightly missing and one still have this question at the end of the Introduction...
2) Discussion. You have indicated starting from Line 425 "Authors should discuss the results and how they can be interpreted from the perspective of previous studies and of the working hypotheses..." Please, make some remarks about your previous studies. You see, when the reader reaches this place, automatically is rised the questions what you mention here. I would like to encourage the authors to write some short paragraph about their previous studies .... I do not insist, but would like to get something more from you here!
3) References. Nice, but are you sure that it is not possible to remove or change these 7 previous century sources? Do not fit for this manuscript!
Author Response
We appreciate all comments and suggestions made by the reviewer, the changes made are in the text marked in yellow. One of the authors, who is native English-speaking, reviewed the manuscript again. Here are our responses to his questions or comments:
Reviewer 2
Dear Authors,
an interesting issue, thank you!
I have just some small remarks:
1) please, give 1-2 sentences in the Introduction (in the beginning) that ELSA and ALEXANDROS can be compared; this is slightly missing and one still have this question at the end of the Introduction.
R. We included a paragraph at the end of the introduction (line 95-100)
" Both countries have longitudinal studies in community-dwelling older people (ELSA and ALEXANDROS), with comparable measurements, similar dates of evaluations and years of follow-up (65,66). Besides, Chile and UK have a Life Expectancy at Birth over 80y and similar Life Expectancy at 65y (UK 20y; Chile 19.9y) (67), but in different cultural and socioeconomic contexts, healthcare systems and demographic dynamics, which makes more valuable the comparison".
2) Discussion. You have indicated starting from Line 425 "Authors should discuss the results and how they can be interpreted from the perspective of previous studies and of the working hypotheses..." Please, make some remarks about your previous studies. You see, when the reader reaches this place, automatically is rised the questions what you mention here. I would like to encourage the authors to write some short paragraph about their previous studies .... I do not insist, but would like to get something more from you here!
R. We appreciate the reviewer's comment, lines 425-428, are part of the instructions included in the template presented by the journal, by mistake it was not deleted. This paragraph was removed from the manuscript.
3) References. Nice, but are you sure that it is not possible to remove or change these 7 previous century sources? Do not fit for this manuscript!
R. Thank you very much for the comment, 5 of them were removed:
- Anstey, K.J.; Smith, G.A. Interrelationships among Biological Markers of Aging, Health, Activity, Acculturation and Cognitive Performance in Late Adulthood. Psychol. Aging 1999, 14, 605–618, doi:10.1037/0882-7974.14.4.605.
- Sliwinski, M.; Lipton, R.B.; Buschke, H.; Stewart, W. The Effects of Preclinical Dementia on Estimates of Normal Cognitive Functioning in Aging. Journals Gerontol. - Ser. B Psychol. Sci. Soc. Sci. 1996, 51, 2–7, doi:10.1093/geronb/51B.4.P217.
-Katz, S.; Ford, A.B.; Moskowitz, R.W.; Jackson, B.A.; Jaffe, M.W. Studies of Illness in the Aged: The Index of ADL: A Standardized Measure of Biological and Psychosocial Function. JAMA J. Am. Med. Assoc. 1963, 185, 914–919, doi:10.1001/jama.1963.03060120024016.
-Radloff, L.S. The CES-D Scale: A Self-Report Depression Scale for Research in the General Population. Appl. Psychol. Meas. 1977, 1, 385–401, doi:10.1177/014662167700100306.
-Camicioli, R.; Howieson, D.; Oken, B.; Sexton, G.; Kaye, J. Motor Slowing Precedes Cognitive Impairment in the Oldest Old. Neurology 1998, 50, 1496–1498, doi:10.1212/WNL.50.5.1496.
Those corresponding to the instruments (GDS-15, CES-D) used were kept (references 70 and 75 in the final manuscript).
- Sheikh, J.I.; Yesavage, J.A. 9/Geriatric Depression Scale (Gds) Recent Evidence and Development of a Shorter Version. Gerontol. 1986, 5, 165–173, doi:10.1300/J018v05n01_09.
- Turvey, C.L.; Wallace, R.B.; Herzog, R. A Revised CES-D Measure of Depressive Symptoms and a DSM-Based Measure of Major Depressive Episodes in the Elderly. Psychogeriatrics 1999, 11, 139–148, doi:10.1017/S1041610299005694.
